**Peer**J

# Population expansions shared among coexisting bacterial lineages are revealed by genetic evidence

Morena Avitia[1], Ana E. Escalante[2], Eria A. Rebollar[1,3], Alejandra Moreno-Letelier[4], Luis E. Eguiarte[1] and Valeria Souza[1]

[1] Departamento de Ecología Evolutiva, Instituto de Ecología, Universidad Nacional Autónoma de México, México DF, México
[2] Departamento de Ecología de la Biodiversidad, Laboratorio Nacional de Ciencias de la Sostenibilidad, Instituto de Ecología, Universidad Nacional Autónoma de México, México DF, México
[3] Biology Department, James Madison University, Harrisonburg VA, USA
[4] Division of Biology, Imperial College London, Silwood Park Campus, Ascot, Berkshire, UK

Corresponding author
Valeria Souza, souza@unam.mx

## ABSTRACT

Comparative population studies can help elucidate the influence of historical events upon current patterns of biodiversity among taxa that coexist in a given geographic area. In particular, comparative assessments derived from population genetics and coalescent theory have been used to investigate population dynamics of bacterial pathogens in order to understand disease epidemics. In contrast, and despite the ecological relevance of non-host associated and naturally occurring bacteria, there is little understanding of the processes determining their diversity. Here we analyzed the patterns of genetic diversity in coexisting populations of three genera of bacteria (*Bacillus, Exiguobacterium,* and *Pseudomonas*) that are abundant in the aquatic systems of the Cuatro Cienegas Basin, Mexico. We tested the hypothesis that a common habitat leaves a signature upon the genetic variation present in bacterial populations, independent of phylogenetic relationships. We used multilocus markers to assess genetic diversity and (1) performed comparative phylogenetic analyses, (2) described the genetic structure of bacterial populations, (3) calculated descriptive parameters of genetic diversity, (4) performed neutrality tests, and (5) conducted coalescent-based historical reconstructions. Our results show a trend of synchronic expansions across most populations independent of both lineage and sampling site. Thus, we provide empirical evidence supporting the analysis of coexisting bacterial lineages in natural environments to advance our understanding of bacterial evolution beyond medical or health-related microbes.

## INTRODUCTION

The present-day distribution of biodiversity is a consequence of the evolutionary dynamics of populations and the history of the regions in which they occur (*Lomolino, Riddle & Brown, 2006*). Lineage diversification results from interactions between the intrinsic

biological constraints of organisms and extrinsic environmental factors (*Dawson, 2012*). The demographic patterns of coexisting populations have often been analyzed in the context of the history of a region to better understand how the evolution of resident lineages has been affected by their environment (*Lessa, Cook & Patton, 2003*; *Carnaval & Moritz, 2008*; *Ramírez-Barahona & Eguiarte, 2013*; *Chan, Schanzenbach & Hickerson, 2014*; *Hope et al., 2014*). The analysis of genetic traits through the application of coalescent theory has provided evidence of the impacts of past geological and environmental events on the demographic history of resident species, such as the effect of Quaternary glaciations upon the distribution of animal species in North America (*Lessa, Cook & Patton, 2003*; *Hope et al., 2014*).

It is clear that patterns of biodiversity across phylogenetically distinct taxa may be influenced by shared historical factors (*Chan, Schanzenbach & Hickerson, 2014*; *Hope et al., 2014*). Despite the importance of microorganisms in ecosystems (*Allison & Martiny, 2008*; *Strickland et al., 2009*), the scale of the impact of historical factors remains poorly understood and little is known about the population dynamics of natural microbial populations, which has greatly hindered our understanding of the processes determining diversity. Most studies on microbial population dynamics have analyzed demographic patterns of bacterial pathogens, undertaken primarily to understand disease epidemics and population expansion events of human pathogens (*Pybus et al., 2001*; *Wirth et al., 2007*; *Tazi et al., 2010*). Studies of demographic trends in natural bacterial populations have been scant (*Guttman, Morgan & Wang, 2008*), due in part to limited sampling of populations from different lineages at similar temporal and geographic scales.

In this context, the Cuatro Cienegas Basin (CCB) harbors high levels of bacterial diversity, arguably due to environmental variation (e.g., salinity) across the aquatic systems within the basin (*Cerritos et al., 2011*). However, there is no conclusive evidence that identifies environmental conditions or geographic distance as predictors for the presence of certain bacterial groups (*Escalante et al., 2008*; *Rebollar et al., 2012*). Thus the influence of shared historical factors in population dynamics could be a plausible explanation for the observed diversity in CCB. A collection of coexisting bacterial isolates from the aquatic systems in this area has been built for over a decade (*Cerritos et al., 2011*; *Rebollar et al., 2012*; *Rodríguez-Verdugo et al., 2012*) and represents a unique opportunity to investigate the historical population patterns of coexisting bacterial lineages in a natural setting.

In the present work, we tested the hypothesis that a shared history in the CCB region has left common genetic signatures across phylogenetically diverse microbial populations. Using population genetics and coalescent-based approaches, we assessed the population history of lineages from two closely related genera of Firmicutes, *Bacillus* and *Exiguobacterium*, as well as lineages of Gammaproteobacteria from the genus *Pseudomonas*. The selected lineages are all common in the natural setting of the CCB aquatic system, and we anticipated that the genetic variation present among coexisting lineages would reveal a signature of common historical dynamics independent of phylogenetic relationships. Multilocus sequence typing (MLST) markers were used to (1) perform comparative phylogenetic analyses, (2) describe the genetic structure of bacterial populations,

(3) calculate descriptive parameters of genetic diversity, (4) perform neutrality tests, and (5) conduct historical demography reconstructions.

Consistent with our hypothesis, we identified a common trend of expansion in the studied populations over a similar time frame that occurred independently of phylogenetic relationships. These results provide empirical evidence that analyzing coexisting bacterial lineages in natural environments can advance our understanding of bacterial evolution, beyond medical or health-related species.

## MATERIALS & METHODS

### Study sites and sampling

Surface water and sediment samples were collected between 2003 and 2009 (*Rodríguez-Verdugo et al., 2012*; *Rebollar et al., 2012*) from four aquatic systems within the CCB: Churince (C), Los Hundidos (H), Mesquites (M), and Pozas Azules (Pa). *Pseudomonas* isolates were collected only from C. All isolates were obtained using previously described methods (*Rebollar et al., 2012*). All samples were collected under the "Vida Silvestre" permit 0531 FAUT-0230 granted by the Mexican government agency: "Comisión Nacional de Áreas Protegidas" (CONANP).

Water and sediment samples were diluted at 1:10; 1:100; 1:1,000; and 1:10,000 using saline solution (1% NaCl). Subsequently, 200 µl of each dilution was plated using either marine agar (Difco 2216) for Firmicutes or GSP agar (*Kielwein, 1971*) for *Pseudomonas*. Firmicutes isolates were incubated for 24 h at 37 °C; *Bacillus* and *Exiguobacterium* colonies were pale or bright orange, respectively. All orange colonies were purified by single-colony isolation. *Pseudomonas* isolates were incubated for 48 h at 30 °C. *Pseudomonas* colonies were selected based on a change in the color of the GSP media to purple. All isolates were stored at −80 °C in 20% (w/v) glycerol.

### Molecular markers and sequencing

DNA was extracted from all isolates using a DNAeasy Tissue Kit (Qiagen, Valencia, CA, USA), following the manufacturer's instructions. The *16S rRNA* gene was amplified and sequenced using 27F and 1492R primers, according to the conditions described by *Lane (1991)*. PCR products were sequenced and used to identify isolates. Once a strain was identified as *Bacillus, Exiguobacterium* or *Pseudomonas*, we amplified and sequenced a set of housekeeping genes commonly used for MLST (*Cerritos et al., 2011*; *Yamamoto et al., 2000*; *Sarkar & Guttman, 2004*; *Rodrigues et al., 2006*). Genus-specific PCR primers (Table S1) were used. All PCR products were sequenced at the University of Washington's High Throughput Genomics Center. Sequences were deposited in the GenBank database with the following accession numbers: *Bacillus citC* (KC900996–KC901178); *Bacillus gltx* (JQ241465–JQ241624); *Bacillus hsp70* (KC901179–KC901361); *Bacillus recA* (JQ247793–JQ247952); *Bacillus spo0A* (KC901362–KC901540); *Exiguobacterium citC* (JF916988–JF917080, JF952020–JF952109); *Exiguobacterium hsp70* (JF952111–JF952292); *Exiguobacterium recA* (JF952293–JF952475); *Exiguobacterium rpoB* (JF952476–JF952658);

*Pseudomonas acnB* (KC953704–KC953753); *Pseudomonas gyrB* (KC920532–KC920576); *Pseudomonas recA* (KC961435–KC961462); *Pseudomonas rpoD* (KC920481–KC920531).

## Phylogenetic reconstructions

The complete sequences of the *16S rRNA* gene and partial sequences of the MLST genes were edited and aligned using BioEdit (*Hall, 1999*). Representative *16S rRNA* sequences for the three genera were obtained from GenBank and were included in the alignments as references (see Figs. S1A–S1C for accession numbers).

We calculated the Pairwise Homoplasy Index ($\Phi_W$) using the "PHI test recombination" function implemented in SplitsTree4 (*Huson & Bryant, 2006*) to verify clonality for all lineages as has been reported for *Bacillus, Exiguobacterium and Pseudomonas* (*Roberts & Cohan, 1995*; *Spiers, Buckling & Rainey, 2000*; *Rebollar et al., 2012*). Since all lineages are clonal (Table S2), we were able to use concatenated data sets for phylogenetic reconstruction and population genetics analyses.

For all genera, maximum likelihood (ML) phylogenies were constructed using (i) *16S rRNA* sequences and (ii) concatenated alignments of MLST sequences. The program jModelTest v.2.1.3 was used to determine the best nucleotide substitution model for each alignment (*Posada, 2008*; *Darriba et al., 2012*). For the *16S rRNA* gene, TPM2uf+I+G, TIM1+I+G and GTR+I+G were the substitution models selected for *Bacillus, Exiguobacterium* and *Pseudomonas*, respectively. GTR+I+G, TIM2+I+G and GTR+I+G were the models selected for the *Bacillus, Exiguobacterium,* and *Pseudomonas* concatenated MLST loci alignments, respectively. Phylogenetic relationships were reconstructed using PhyML v.3.0 (*Guindon & Gascuel, 2003*), with the corresponding DNA substitution models, tree improvement was carried out by Subtree Pruning and Regrafting, and branch support was evaluated by 1,000 bootstrap pseudo-replicates.

## Population structure

We defined populations as the minimal study unit in our collection. Populations have to be defined considering both geographic and genetic criteria. Given that it is possible to find populations (genetic pools) that are not site-restricted we investigated population structure by genetic differentiation, which is defined as changes in the frequency distribution of haplotype variants among subpopulations (*Hartl & Clark, 1997*). We estimated pairwise $F_{ST}$ values among sampling sites within lineages. With this approach we defined a single population as all isolates from a lineage that exhibited no significant genetic differentiation as measured by pairwise $F_{ST}$ estimates. Pairwise $F_{ST}$ estimates were obtained for each monophyletic lineage across all sampling sites using Arlequin 3.5 (*Excoffier & Lischer, 2010*), with 1,000 iterations. For the *Pseudomonas* isolates, $F_{ST}$ analysis was not performed since all isolates were from the same sampling site.

The $F_{ST}$ approach implicitly incorporates geographic location and genetic criteria in the investigation of population structure. However, it is possible that genetic structure exists for each lineage when looking at individual genetic variants within each population. Thus, we investigated potential substructure within the groups defined by $F_{ST}$ by taking a Bayesian approach implemented in BAPS 6 (*Tang et al., 2009*). The Bayesian analysis of

population structure was performed using the option for linked loci, specifically developed for MLST data (*Tang et al., 2009*). A maximum number of clusters (K) was set to ten, or equal to the number of individuals if these were fewer than ten. Each analysis was replicated ten times.

Once populations were defined, standard measures of nucleotide diversity ($\pi$) and the mutation parameter Watterson's $\theta$ were estimated, together with neutrality tests (Tajima's *D*, Fu and Li's F* and D* and Fu's $F_S$) using DNAsp v4.1 (*Rozas et al., 2003*). Details of these calculations can be found in the Supplemental Information.

### Historical population dynamics

We performed an Extended Bayesian Skyline Plot (EBSP) analysis as implemented in Beast v.1.7.5 (*Drummond & Rambaut, 2007*; *Heled & Drummond, 2008*). The EBSP infers past population dynamics from a sample of contemporary sequences, taking into account the genealogical stochasticity of the coalescent (*Ho & Shapiro, 2011*; *Drummond et al., 2005*). Additionally, this method does not depend on a specific *a priori* demographic model, but infers the number of population size changes directly from the data. As a result, it provides a measure of statistical credibility of the inferred number of population size changes compared to the alternative of constant population size (*Heled & Drummond, 2008*).

For the EBSP analysis, we used all MLST genes, a strict molecular clock, and the same nucleotide substitution models used for phylogenetic reconstruction to estimate changes in population size for each genetically homogeneous population. The substitution rate used for the two Firmicutes genera (*Bacillus* and *Exiguobacterium*) was $7 \times 10^{-9}$ substitutions/per site/generation, obtained from a dated phylogenetic tree (*Maughan, 2007*). The substitution rate used for *Pseudomonas* was $2.8 \times 10^{-8}$ substitutions/per site/generation, based on an experimental evolution study of *P. fluorescens* (*Barrett, MacLean & Bell, 2006*). All time estimates obtained were expressed in number of generations. Changes in population size were expressed as a function of the product of $N_e$ and the generation time ($N_e * t$). All analyses were run for 50–100 million generations, until adequate mixing was achieved. Ten percent of burn-in was removed and the sampling was done every 1,000 chains. The rest of the parameters were set according to the guidelines of *Heled & Drummond (2008)*. Results were analyzed with Tracer v.1.5 and LogCombiner v.1.7.5 (*Drummond & Rambaut, 2007*).

## RESULTS

To investigate the influence of a common habitat in the genetic variation of bacterial populations, we analyzed MLST data from a collection of isolates across three bacterial genera (*Bacillus*, *Exiguobacterium* and *Pseudomonas*) sampled from water bodies in the geographic region of CCB, Mexico.

### The phylogenetic history of CCB lineages

Phylogenetic reconstructions based on MLST and *16S rRNA* sequences were used to assign isolates to monophyletic lineages (Figs. 1 and S1 respectively). MLST phylogenies

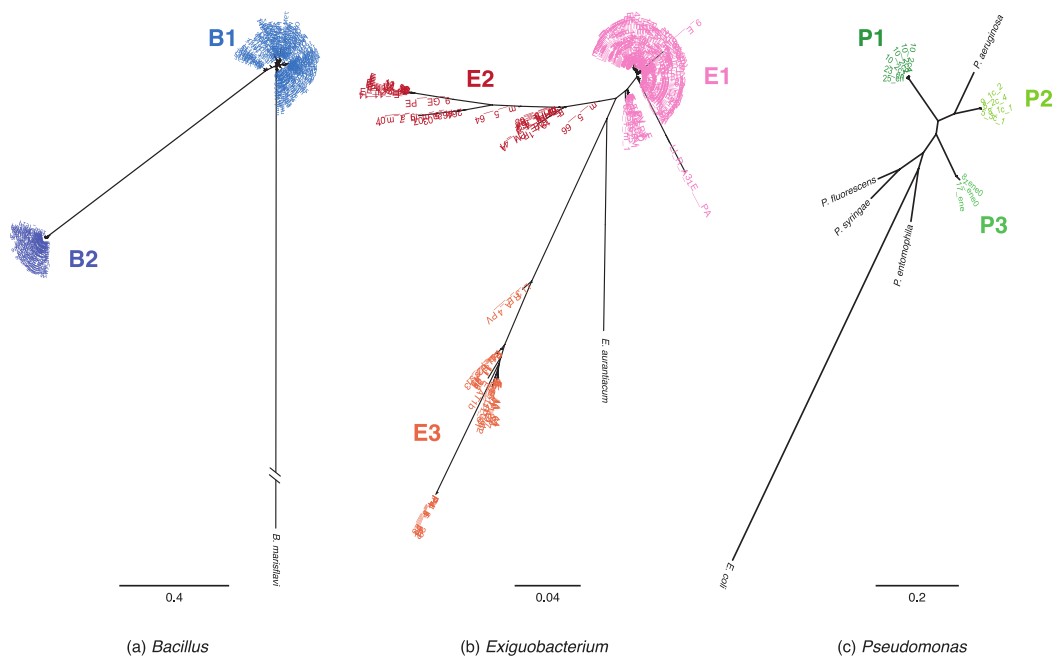

| (a) *Bacillus* | (b) *Exiguobacterium* | (c) *Pseudomonas* |

**Figure 1** **MLST phylogenetic reconstructions of isolates collected in the Cuatro Cienegas Basin (CCB), Mexico.** Phylogenies were constructed using the maximum likelihood approach and are based on concatenated sequences of at least four MLST loci. (A) *Bacillus* lineages with *B. marisflavi* as outgroup, and *citC, gltX, hsp70, recA* and *spo0A* loci, (B) *Exiguobacterium* lineages with *E. auranticum* as outgroup, and *citC, hsp70, recA* and *rpoB* loci, and (C) *Pseudomonas* lineages with *Escherichia coli* as outgroup, and *acnB, gyrB, recA* and *rpoD* loci. The scale of the bar represents the number of substitutions per site.

identified at least two lineages within each genus. We named the lineages according to genus: "B" for *Bacillus*, "E" for *Exiguobacterium*, and "P" for *Pseudomonas*.

Among *Bacillus* isolates, we identified two clusters, B1 and B2 (Fig. 1A); their closest relatives were *B. aquimaris*, *B. vietnamensis*, *B. marisflavi*, and *B. coahuilensis* (Fig. S1A). *Exiguobacterium* isolates clustered into three well-defined groups: E1, E2, and E3 (Fig. 1B). For clusters E1 and E2, the most closely related species was *E. aurantiacum*, and *E. profundum* was the closest relative of cluster E3. *Pseudomonas* isolates were divided into three groups: P1, P2, and P3 (Fig. 1C). P1 included the majority of isolates and, according to *16S rRNA* sequences, was closely related to *P. oryzihabitans*. P2 isolates were closely related to *P. otitidis* and *P. aeruginosa*, while P3 had recently been described as a new species, *P. cuatrocienegasensis* (*Escalante et al., 2009*) (Fig. S1C).

## Population structure
Populations defined by pairwise $F_{ST}$ values corresponded with sampling-site pairs with non-significant $F_{ST}$ values (Table S3). We named the resulting populations indicating lineage and sampling site as: B1_C, B1_HPa, B1_M, B2_C, B2_MPa, E1_CHPa, E1_M, E2_CH, E2_M, E3_CM, P1_C, P2_C, and P3_C. Further exploration of structure within lineages was performed with BAPS and showed clusters that are shared among sites within lineages. In all cases, differences in frequency distribution of haplotypes (clusters) correspond with the population structure that we identified through pairwise $F_{ST}$ (Fig. 2).
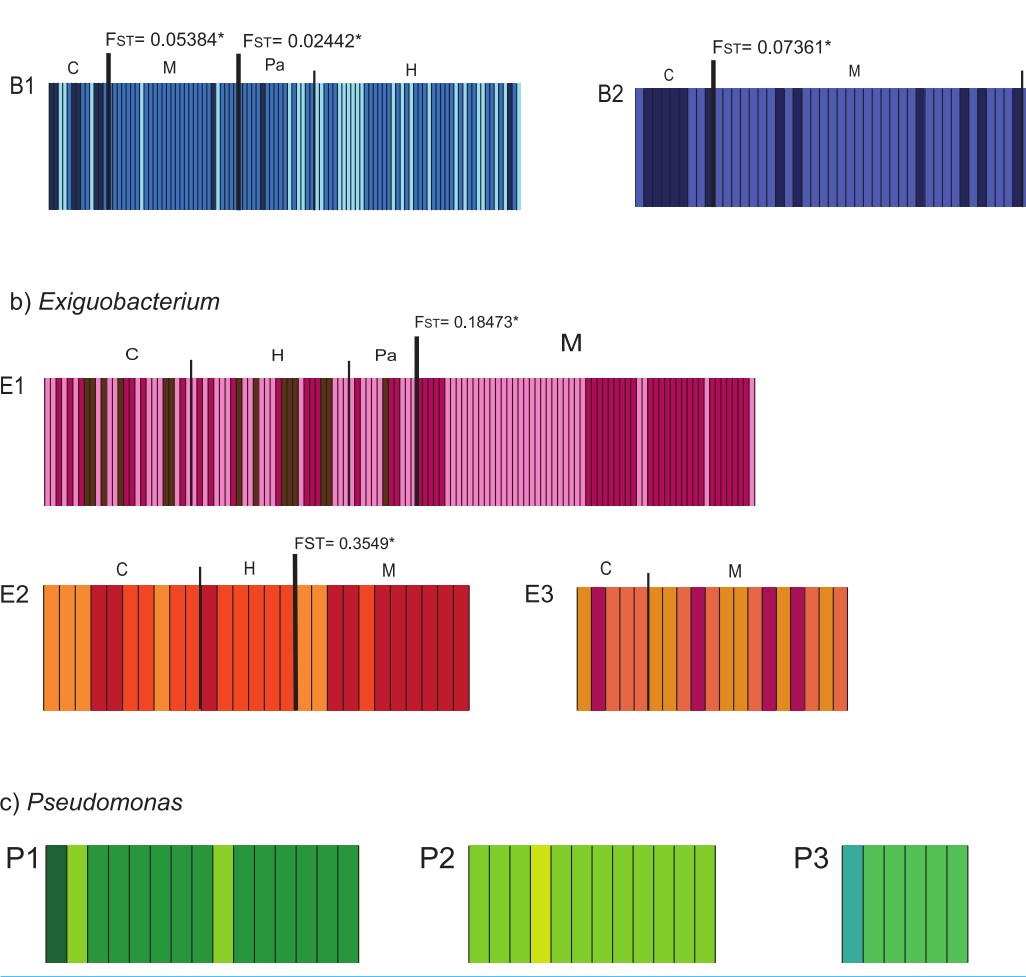

**Figure 2** **Bayesian Analysis of Population Structure (BAPS) of isolates collected in the Cuatro Cienegas Basin (CCB), Mexico.** The analysis was conducted for each studied lineage within each genus (*Bacillus* = B1, B2; *Exiguobacterium* = E1, E2, E3; *Pseudomonas* = P1, P2, P3) using the option for linked loci. The maximum number of clusters (K) was set to 10, or equal to the number of individuals if these were fewer than 10. Each analysis was replicated 10 times. The columns represent a single multilocus genotype and are identified with distinctive colors corresponding to different genetic clusters. Different populations are graphically denoted with a dark and wide line between populations with significant $F_{ST}$ values, which are also shown.

Therefore, we delimited populations based on $F_{ST}$ estimates since it incorporates both the geographic and genetic components in the definition of populations.

## Population history: population genetics and coalescent approaches

Results of the genetic diversity analyses and neutrality tests (including Fu's $F_s$; *Fu, 1997*) are presented in Table S4. Neutrality tests suggested expansion events but were inconclusive. In our data set, most values for Tajima's D and Fu & Li's tests were negative, although non-significant, which is suggestive of expansion events of populations. However, Fu's $F_s$,

a test that was explicitly designed to evaluate population expansions, was not significant in most cases. Overall, these classic population genetics estimates did not allow for strong conclusions about historical population events.

To further explore the influence of historical processes in shaping the diversity of these bacterial populations, we used a coalescent approach. Extended Bayesian Skyline Plot (EBSP) analyses of populations (genetic pools) showed signals of expansion in 9 of the 13 populations, of which 4 have statistical support for at least one change in population size (Fig. 3; for statistical support values of the population size changes see Table S4). Within *Bacillus,* an expansion (although not statistically supported) was detected in one population (B1_HPa) that included most B1 isolates. Plots for the other two B1 populations (B1_M and B1_C) showed constant population size across generations (Fig. 3). The B2 plots showed statistical support for change in population size in at least one population (B2_MPa) and constant population size in another (B2_C). Within *Exiguobacterium*, there was evidence of expansion in one of two E1 populations (E1_M), in one of the two E2 populations (E2_CH) and a trend of expansion (not statistically supported) in the only E3 population (E3_CM). Finally, the plots for all *Pseudomonas* populations displayed signals of expansion, although only P2_C was statistically supported. Overall, a trend of expansion was observed for the studied populations independent of phylogenetic relationships or sampling sites. Moreover, the skyline plots revealed a common time frame for the start of expansions (20,000 to 30,000 generations ago, Fig. 3), regardless of lineage- or locus-specific parameters.

## DISCUSSION

Diversity patterns in nature are a consequence of the evolutionary dynamics of populations and the history of the regions in which populations occur (*Lomolino, Riddle & Brown, 2006*). A powerful approach to unveil the influence of historical factors is to analyze current genetic variation to reconstruct the demographic history of populations (*Lessa, Cook & Patton, 2003*; *Hope et al., 2014*). In microorganisms, genetic studies looking at historical demography have mainly focused on host-associated bacteria to gain insight in disease epidemics (*Mhedbi-Hajri et al., 2013*; *Wirth et al., 2006*; *Comas et al., 2013*; *Holt et al., 2013*). However, little is known about population dynamics of non-host associated and naturally occurring bacteria.

When analyzing bacterial populations caution should be taken in the delimitation of populations, which has to consider key aspects such as the degree of genetic and ecological diversity. It is also important to consider the molecular resolution at which populations are defined (*Kopac & Cohan, 2011*). The more variable loci studied, the more populations may be identified. In our case, few and highly conserved loci may result in a coarse population delimitation. Nonetheless, to include both genetic and ecological aspects in the delimitation of populations, we defined individual populations for each lineage by analyzing genetic structure across sampling sites. The analysis showed that populations defined by the $F_{ST}$ statistic include different sampling sites with contrasting environmental conditions (*Rebollar et al., 2012*), which is in accordance with previous work where no

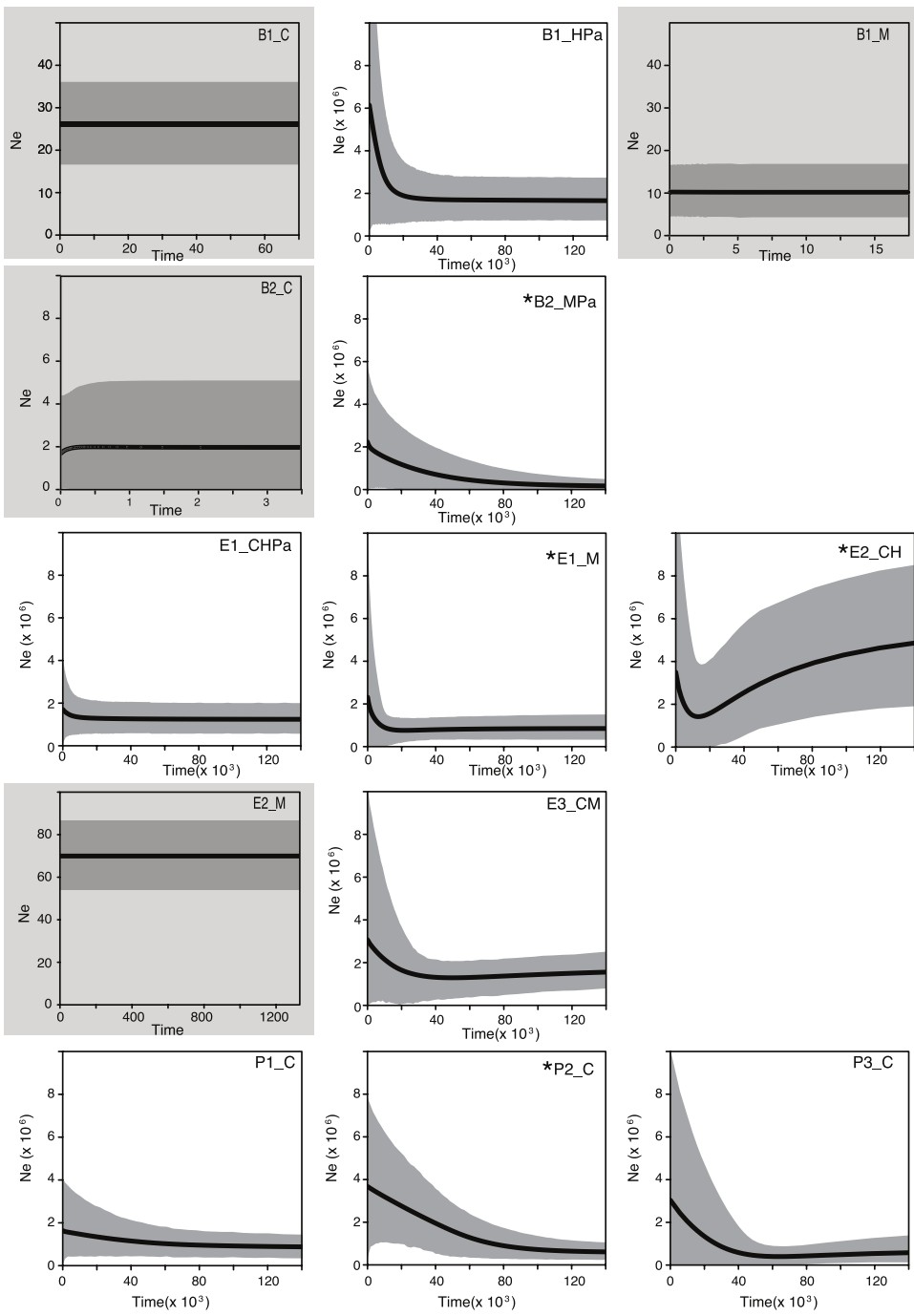

**Figure 3 Bayesian skyline plots for lineages collected in the Cuatro Cienegas Basin (CCB), Mexico.** Showing the effective population size through time (black line). The shaded area represents 95% credibility intervals. Labels denote the lineage and sampling sites of each population. B, E, and P stand for *Bacillus*, *Exiguobacterium*, and *Pseudomonas*, respectively. C, H, M, and Pa stand for the sampling sites Churince, Los Hundidos, Mesquites, and Pozas Azules, respectively. Shaded plots represent populations with constant population sizes. Asterisks indicate populations with statistically supported demographic changes (Table S5).

evidence of geographic or environmental barriers to dispersal exists (*Escalante et al., 2008*; *Rebollar et al., 2012*). We explored the existence of subpopulations within the $F_{ST}$-defined populations following a Bayesian approach implemented in BAPS (*Tang et al., 2009*), and we found more clusters that subdivided even further the populations defined with $F_{ST}$ estimates. However, genetic diversity within the identified populations ($\pi = 0.00032$–$0.04698$ and Watterson's $\theta = 0.00073$–$0.047$; Table S4) is in the range of values reported for natural bacterial populations (*Roberts & Cohan, 1995*; *Vos & Velicer, 2006*), meaning that the observed diversity is within the range of normal polymorphism in bacterial populations. Overall, $F_{ST}$, BAPS, and diversity estimates support our population delimitation and indicate that we are not including unusually divergent individuals into the studied populations which might confound the population genetics and expansion estimations. Given that classic population genetics estimates did not allow for strong conclusions about historical population events, we used a coalescent approach to further explore the influence of historical processes. From this approach, we found evidence showing that common historical events were population expansions that occurred in 9 out of 13 populations (Fig. 3). Even more, it is noteworthy that at least one population of each genus presented evidence of expansion that was statistically supported. In clonal and free-living populations of bacteria, evolutionary models predict that populations experience strong fluctuations, either by selective sweeps or by metapopulation dynamics (*Fraser et al., 2009*; *Kopac & Cohan, 2011*; *Shapiro & Polz, 2014*). The results obtained in this study can be explained by these models since the analyzed lineages come from a natural setting and present no evidence of recombination (Table S2; *Roberts & Cohan, 1995*; *Spiers, Buckling & Rainey, 2000*; *Rebollar et al., 2012*). Without recombination, clonal clusters emerge by random mutation, but can quickly drift to extinction unless they have a major selective advantage, in which case their presence could suggest the emergence of new ecologically distinct populations (*Shapiro & Polz, 2014*) or ecotypes (*Kopac & Cohan, 2011*). Nonetheless, given that we do not have conclusive evidence that identifies environmental conditions or geographic distance as predictors for the presence of certain bacterial groups or populations (*Escalante et al., 2008*; *Rebollar et al., 2012*), the influence of shared historical factors in population dynamics could be a plausible explanation for the observed population expansion events in CCB. Given that we used just a few loci to delimit the studied populations, it should be noted here that we cannot rule out the possibility that the observed expansions may be a reflection of the emergence of closely related populations (undetectable to our molecular markers), and not population expansions in size. This ambiguity could be resolved if more variable loci, even complete genomes, are analyzed. Despite the alternative interpretations of the results, we consider notable the fact that we identified the same type of events (either diversification or population expansions) in most populations in the same time frame.

Population expansions in our data were identified to occur during the same time frame ($\cong 20,000$ generations ago). Our estimation of time to the most recent population change is calculated in number of generations. Although generation times may vary considerably among bacterial lineages (*Mason, 1934*; *Errington, Powell & Thompson,*

*1965*), laboratory observations of growth rates of the lineages included in this study do not suggest major differences in generation times between them, which supports the synchrony argument. In this study, we used previously reported estimates for Firmicutes (*Bacillus* and *Exiguobacterium*) and *Pseudomonas* to parameterize the coalescent simulations. We used the same substitution rate for *Bacillus* and *Exiguobacterium* since little information is available for *Exiguobacterium* and also because *Maughan (2007)* did not detect a considerable rate of heterogeneity between sporulating and non-sporulating Firmicutes. In this respect, it should be acknowledged that values of substitution rates are critical for the estimation of the temporal scale for the demographic events that result from ESPB analyses (*Ho et al., 2011*) and ideally should be directly estimated for the studied sample. We are aware that this could impact the temporal scales of our skyline plots, and be of particular relevance when comparing lineages with different substitution rates, but we do not have any reason to doubt the substitution rates reported in the literature. Further work should try to directly estimate substitution rates for the studied lineages to minimize the possibility of different time scales for the expansion events.

Despite the fact that expansions are expected in bacterial populations, the time of such expansions may differ among lineages because they depend on specific traits such as mutation rate, life history, and population structure (*Avise, 2000*; *Avise, 2009*; *Kuo & Ochman, 2009*; *Maughan, 2007*; *Woolfit & Bromham, 2003*). The synchrony of expansions in coexisting populations of different lineages are not expected to be observed unless shared environmental and historical factors have a major influence on population dynamics obscuring evidence of lineage specific adaptation, as has been observed in different organisms including "human associated" pathogens (*Ornelas et al., 2013*; *Comas et al., 2013*; *Falush et al., 2003*). The synchrony of population expansions supports the argument that current genetic variation may be the result of a major regional event over all populations.

Currently, we do not have the necessary information or samples from the appropriate temporal and spatial scales to determine the environmental changes associated with the historical events that influenced population dynamics of the studied lineages. However, we can propose an approximate timespan in which these demographic events occurred and hypothesize on the environmental causes behind our observations. Most estimates of generation times for bacteria have been determined in laboratory conditions, which may be considerably shorter than generation times in the wild (for *Escherichia coli* see *Pierucci, 1972*; *Ochman, Elwyn & Moran, 1999*). Thus, we estimated the approximate absolute time of population expansions based on generation times for aquatic bacteria estimated in conditions that are similar to their natural habitat conditions (approximately between 75 and 300 generations per year; *Jannasch, 1969*; *Hendricks , 1972*). We selected a generation time at the lower limit (75 gen/yr), because it is possible that the bacteria in our study may grow slowly, given the extreme limitation of nutrients, in particular phosphorous, found in the CCB aquatic system (*Souza et al., 2006*; *Souza et al., 2012*; *Souza et al., 2008*). Thus, if the synchronic event occurred 200–300 years ago for most populations, it is tempting to suggest that the population expansions we observed reflect changes resulting from the

recent human activity in the CCB area. In particular, an increase in anthropogenic activity during the late eighteenth and nineteenth centuries has been reported, primarily centered around agriculture and ranching (*Alessio-Robles, 1946*) which coincides with the time frame of expansions found here.

## CONCLUSIONS

This study investigates historical population events of coexisting but phylogenetically distant bacterial lineages. To our knowledge, this is the first report of a synchronous signature of population expansion that occurred independently of phylogenetic relationships, which is in accordance with our initial hypothesis. Our findings provide strong evidence for the potential of comparative population genetics to address questions about the influence of shared historical factors in the population evolutionary processes of naturally occurring non-host associated bacteria. This information may be important for natural resource management in the context of the ecosystem services that microorganisms provide.

## ACKNOWLEDGEMENTS

We thank Dr. Evan Carson, Dr. Michael Travisano, Dr. Daniel Piñero, Dr. Juan Pablo Jaramillo and Dr. Santiago Ramírez-Barahona for their constructive reviews of the manuscript. We also thank Laura Espinosa, Dr. Erika Aguirre and Jaime Gasca-Pineda for technical support during the development of the project.

This paper constitutes part of the doctoral research of the first author, who thanks the Doctorado en Ciencias Biomédicas, Universidad Nacional Autónoma de México (UNAM) and the Consejo Nacional de Ciencia y Tecnología (CONACyT).

### Funding

The first author received a scholarship and financial support from the Consejo Nacional de Ciencia y Tecnología (CONACyT, grant no. 210335). This work constitutes part of the doctoral research of the frist author who received scholarship and financial support from the Consejo Nacional de Ciencia y Tecnologa (CONACyT, grant no. 210335). The project was supported by a grant from WWF-Alianza Carlos Slim. The funders had no role in study design, data collection and analysis, decision to publish, or preparation of the manuscript.

### Grant Disclosures

The following grant information was disclosed by the authors:
Consejo Nacional de Ciencia y Tecnología (CONACyT): 210335.
WWF-Alianza Carlos Slim.

### Competing Interests

Valeria Souza and Luis E. Eguiarte are Academic Editors for PeerJ.

## Author Contributions

- Morena Avitia, Ana E. Escalante and Eria A. Rebollar conceived and designed the experiments, performed the experiments, analyzed the data, wrote the paper, prepared figures and/or tables, reviewed drafts of the paper.
- Alejandra Moreno-Letelier conceived and designed the experiments, analyzed the data, contributed reagents/materials/analysis tools, wrote the paper, prepared figures and/or tables, reviewed drafts of the paper.
- Luis E. Eguiarte and Valeria Souza conceived and designed the experiments, contributed reagents/materials/analysis tools, reviewed drafts of the paper.

## Field Study Permissions

The following information was supplied relating to field study approvals (i.e., approving body and any reference numbers):

All samples were collected under the "Vida Silvestre" permit 0531 FAUT-0230 granted by the Mexican government agency: "Comisión Nacional de Áreas Protegidas" (CONANP).

## DNA Deposition

The following information was supplied regarding the deposition of DNA sequences:

Sequences were deposited in the GenBank database with the following accession numbers: *Bacillus citC* (KC900996–KC901178), *Bacillus gltx* (JQ241465–JQ241624), *Bacillus hsp70* (KC901179–KC901361), *Bacillus recA* (JQ247793–JQ247952), *Bacillus spo0A* (KC901362–KC901540), *Exiguobacterium citC* (JF916988–JF917080, JF952020–JF952109), *Exiguobacterium hsp70* (JF952111–JF952292), *Exiguobacterium recA* (JF952293–JF952475), *Exiguobacterium rpoB* (JF952476–JF952658), *Pseudomonas acnB* (KC953704–KC953753), *Pseudomonas gyrB* (KC920532–KC920576), *Pseudomonas recA* (KC961435–KC961462), *Pseudomonas rpoD* (KC920481–KC920531).

## Supplemental Information

Supplemental information for this article can be found online at http://dx.doi.org/10.7717/peerj.696#supplemental-information.

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
