# Peer review of "Population expansions shared among coexisting bacterial lineages are revealed by genetic evidence"

_PeerJ, doi:10.7717/peerj.696_

## Round 0.1 · original submission · Major Revisions

· Academic Editor

Major Revisions

Your paper was reviewed by one ad hoc reviewer and by me, as I was not able to obtain a second ad hoc review.

Reviewer 1 had some very significant comments addressing the accuracy of the time estimates for the demographic expansions, noting that the estimates for both mutation rate and the number of generations may have large errors. A revised paper will need to deal with the confidence limits of the estimates.

The reviewer suggested a more detailed analysis of the population genetic results.

The reviewer also suggested some important style improvements that you should consider, for example, eliminating repetitive paragraphs.

My own primary concern is that the EBSP approach is likely not appropriate for your data set. As laid out by Ho and Shapiro in their 2011 paper (which you've cited), it is important that a data set to be analyzed include only one population. They demand that a population be panmictic but I would say that in the context of rarely sexual organisms, the focus group should be one population of ecologically interchangeable organisms. Ho and Shapiro suggest that if there is doubt about the homogeneity of a possible focus group, the researcher should instead analyze subsets that are likely to each represent single populations. I'll note that Heller et al. (Heller R, Chikhi L, Siegismund HR. 2013. The confounding effect of population structure on Bayesian skyline plot inferences of demographic history. PLoS One 8:e62992.) point out that a structured population will likely give false evidence of a population expansion. To my mind, it seems that your study is more likely to have shown an expansion in the number of populations than expansion in the size of a given population. On the other hand, the demographic expansion you have seen might instead be a reflection of the presence of a large number of closely related populations, which may not have increased in number at all.

Related to this, I appreciate that you have attempted to analyze small phylogenetic clusters within each genus, but I think you'll need to subsample phylogenetically to see if the evidence for demographic expansion holds up. You'll still need to deal with the possibility that even your small phylogenetic groups might be heterogeneous, that is, multiple populations.

I would also like to suggest some less significant points (in my role as reviewer).

You should have explained much earlier that you were focusing on small clades within each genus.

You haven't said much about the ecology of CCB. For example, it shouldn't have come as a surprise that you were isolating bacteria using saline solution.

Related to the problem of analyzing groups that may have been heterogeneous, you should take into account that the population genetic estimates are only valid for single populations. I've written about this in my chapter with Sarah Kopac (A Theory-Based Pragmatism ....).

In the last paragraph of the Results, you should explain more fully the different kinds of results you obtained.

Re line 283, I'd like to offer that Bacillus dispersal is not likely limited, owing to their spore phase.

Reviewer 1 ·

Basic reporting

This manuscript presents a comparison of the demographic histories of several bacterial taxa. The approach is certainly an interesting one, although there are many sources of uncertainty. The authors have taken an appropriately cautious approach. The analysis has value but I have a few suggestions for improvement.

I found a moderate amount of repetition throughout the manuscript. In particular, the first two paragraphs of the Discussion mostly repeat points already made in the Introduction. Please consider deleting these paragraphs after merging any new points into either the Introduction or the remainder of the Discussion. On the other hand, I did not see much discussion of the results of the population genetics analyses. Please provide some interpretation of these results.

Experimental design

The analysis has value but I have two major suggestions for improvement.

(1) An important parameter in the analysis is the mutation rate. If the mutation rates are incorrect, then the scales of the plots cannot be compared with each other. So I think there needs to be a much more critical consideration of the mutation rates used in the analyses.

(2) Line 255 “Figure 2 shows only the skyline plots in which demographic signals were observed” – what happened in the remaining 4 analyses? All of the EBSP analyses should have produced some sort of plot. I am not sure what the authors mean by “significant” (Figure 2 caption - line 500). Generally the interpretation of the skyline plots is not convincing – the uncertainty in some of the plots is so large that we cannot be sure there is a genuine demographic expansion. Can the authors test these patterns in the skyline plots, maybe using Bayes factors?

Validity of the findings

The suggestions about possible causes of the apparent demographic expansions are extremely speculative, especially because of the uncertainty in the generation times. However, the authors seem to have provided sufficient caution with their interpretations. I think that a similar discussion of the choice of mutation rate would be helpful (see comment #1 above).

Additional comments

Minor comments

Lines 27-28. I’m not sure what is meant by “biodiversity” patterns in this context. The first sentence of the abstract can easily be deleted without negative impact.

Line 44. Delete “further” from this sentence because it is redundant.

Lines 55-56 “events in a geographic area impact the demographic history of resident species”. This seems pretty much self-evident. I think that it would be better to make a more meaningful statement about the significance of the findings from coalescent-based analyses, for example “the application of coalescent theory has provided evidence of the impacts of past geological and environmental events on the demographic history of resident species …”

Line 86. Skyline plots are not based on Approximate Bayesian Computation. They are based on Bayesian phylogenetic analysis.

Line 195. Replace “de influence” with “the influence”.

Line 245. Delete comma after “neutrality”.

Line 271. Replace with “Comparative genetic studies of coexisting populations can …”

Line 276. Replace “focused in” with “focused on”.

Line 277. Replace “insight in” with “insight into”.

Line 289. Replace “synchronicity” with “synchrony”, here and throughout the manuscript.

Line 291. Replace “factors impose a major influence in” with “factors have a major influence on”.

In the Figure 1 caption, explain what the scale bars represent.

---

## Round 0.2 · Minor Revisions

· Academic Editor

Minor Revisions

You have made many significant improvements in the manuscript. I was especially moved by your careful analyses to identify your populations. I am still not certain you have found the most newly-divergent populations, as the populations may be splitting at a rate higher than can be resolved with a few loci. I think you should mention molecular resolution as being a possible issue in correctly identifying the populations.

You should be careful to follow Reviewer 1's advice about how to report the outcome of Bayesian analyses, avoiding the concept of significance.

I agree with Francisco Moore's suggestion that you are presenting more analyses than you need to make your point, and that the reader may become a bit dizzy trying to tease out the major conclusions from your analyses. So, please consider what analyses you could downplay or exclude to make your results section punchier.

This reviewer has questioned whether you had strong evidence for population expansions, addressing first an issue with multiple comparisons and then a sign-test issue, that 9/13 is not much different from 1/2. However, my understanding is that each of your cases for population expansion was compelling in itself, based on the skyline analysis. I would like you to address these issues.

Please address all the various suggestions laid out by the reviewers.

Reviewer 1 ·

Basic reporting

The authors have addressed the concerns raised in my previous review. My main comment now is that they need to be careful with the description of the EBSP results (specifically, they should avoid using the term “significant” to describe the results of a Bayesian analysis).

Line 20. Delete comma after “theory”.
Lines 48-49. The authors might wish to note that there has been recent work in this area (Chan et al. 2014; Hope et al. 2014). The topic of synchronous population expansions, which is mentioned by the authors at several points in their manuscript, was investigated in both of these studies.
Lines 61 and 324. Replace “correlates” with “identifies”?
Line 69. Replace “at the CCB region” with “in the CCB region”.
Lines 96 and 98. Replace “during” with “for”.
Line 124. Replace “Splits Tree4” with “SplitsTree4”.
Line 133. Replace “selected for the” with “selected for”.
Line 136. Please provide a bit more detail about the settings used in PhyML.
Line 161 and elsewhere. “Waterson” should be “Watterson”.
Line 172. Replace “outermost branches” with “branches towards the tips”?
Line 179. Missing closing parenthesis after “EBSP”.
Lines 186-187. The terms “significance” and “confidence interval” apply to frequentist statistics, not to the Bayesian approach (this needs to be corrected throughout the manuscript). I think that this sentence can be deleted (the content is implicit in the interpretation of the results, described later in the manuscript).
Lines 196-197. Please give the sampling frequency and proportion of samples discarded as burn-in (if any).
Line 216. Replace “oryzhabitans” with “oryzihabitans”.
Lines 245-247. This sentence does not seem to be correctly constructed.
Line 262. “statistically significant” should not be used when describing the results of a Bayesian analysis (here and elsewhere). It would be better to state the details of the results, for example “the 95% credibility interval of the number of population-size changes did not exclude a value of zero” or “there was support for at least one change in population size”.
Line 274. Replace “lineage or locus-specific” with “lineage- or locus-specific”.
Line 284. Replace “focused in” with “focused on”.
Line 352. Hyphenate “human associated”.
Lines 358-359. Replace “we can approximate a time span” with “we can propose an approximate time span”.
Line 363. Replace “we approximated the absolute time” with “we estimated the approximate absolute time”.
Line 372. Replace “eighteen and nineteen centuries” with “eighteenth and nineteenth centuries”.
Line 572. Replace “dark and wide line” with “thick dark line”.
Line 575. Replace “confidence intervals” with “credibility intervals”.
Line 578. The shading mentioned here (plots with constant population sizes) is quite difficult to see in the Figure. I suggest an alternative method for indicating the plots displaying constant population sizes.

References

Chan, Y. L., Schanzenbach, D., and Hickerson, M. J. (2014) Detecting concerted demographic response across community assemblages using hierarchical approximate Bayesian computation. Molecular Biology and Evolution, 31: 2501-2515.

Hope, A., Ho, S. Y. W., Malaney, J. L., Cook, J. A., and Talbot, S. L. (2014) Accounting for rate variation among lineages in comparative demographic analyses. Evolution, in press.

Experimental design

No further comments.

Validity of the findings

No further comments.

Additional comments

No further comments.

·

Basic reporting

The acknowledgements include a reference to a scholarship that funded the first author, which should likely be included only in the funding statement. Aside from that very minor point, I detected no deficiencies that bear mentioning.

Experimental design

The research described is remarkably rigorous, and shows a high level of care on the part of the researchers. The methods were clear and detailed. The research design was quite logical.

Validity of the findings

The findings are robust and very interesting. I could detect no reasons to doubt their validity. The conclusions flow logically from the data, and the authors are commendably restrained in both their conclusions and acknowledgement of the limitations of the study. Moreover, this paper is clearly a first step, and they acknowledge this.

Additional comments

I find this to be uniformly excellent and well-written manuscript. The authors are to be commended for their outstanding work. I think this paper well warrants publication.

In the absence of any apparent flaws to comment upon, I will note that the comma on line 20 is unnecessary and should be deleted.

·

Basic reporting

No Comments

Experimental design

No Comments

Validity of the findings

In the results (line 240) you mention 7 of 13 lineages had significant Tajima’s D at at least one locus. In context this is interesting but somewhat less convincing since about 3 of those 13 would have had a significant results randomly given that there are 65 total tests.
In the discussion (lines 259 to 274), the EBSPs the trend may have 9 of 13 trending toward expansion but this is not significantly different from a 50:50 probablility of going either way (exact p value for 9 or more in the same direction is 0.1334). The four significant expansions are noteworthy, as you would expect that only one is reasonable expected at most.

Additional comments

This paper is interesting in its subject and is meticulous and detailed in its description of the population and phylo genetics of the system. I think that the improvements from the previous version to the present version are substantial. I certainly think that the material aspects (i.e. data, and basic analysis) are acceptable for publication, but I think that some care in the interpretation of the data would be helpful.

The multifaceted analysis demonstrates the rigor the author’s have applied to ensuring that they have correctly analyzed a complex data set. I am confident they have tried to really explore the data. I have to admit to being a little worn out on the profusion of population genetic and phylogenetic methods, which each have their adherents, to which authors are expected to pander. Each may have a nuance that makes it a bit better for one analysis or another and some can be combined to give a more refined understanding. In the end I find that a common approach is to include a wide diversity of measures and then talk about what you want to while ignoring the clutter. I understand this as a defense, even if I think it detracts from the message as a whole at times. I feel that this manuscript suffers from inclusion of more analysis than is needed. I would narrow the main table down to the few measures that are really needed and include others in supplemental materials if they are not extensively reviewed.

The difficulty with this manuscript is that it is utilizing numerous lines of investigation, each of which may lack inferential power to build a general picture of what is going on. I think it is necessary in this type of study and I have taken a similar approach at times. However, because that approach is necessarily complex and open to attack at each stage it is important to keep out as much clutter as possible. I see focused inference based on Fs, Fst/BAPs clustering, phylogenetic reconstruction and skyline plots. The other material just adds confusion to the story. If they are pointers you used to lead you to the other methods throw then in supplemental materials, and include a line that says there results indicated that the direction of the other methods.

There is a reason I say that you should drop all that material although it may have brought you in a direction. That is because I think that the conclusions of this paper will not be simple and clear. You have good data, it gives you a partial picture and you need that to be as clear as possible. In my opinion I see and respect your interpretation. I am not confident that interpretation is correct, but the more clear you are on where your interpretation comes from the better the reader can judge. I don’t have to agree, I just need to make sure the reader understands.

This leads me to the only substantive critique of the manuscript. I think this walks a fine line of over interpretation. While I think it is on the good side of the line I would rather the interpretation was a little more conservative. This means I would prefer to see a more softly worded set of interpretations. I think your data hints at what you are seeing but does not warrant a clear conclusion. I think a wise reader will see this, but I think it is better to point out the weakness of the inference. This is a general comment I realize and I would not insist on specific changes but I would feel more comfortable with a bit more circumspect interpretation. This is most conspicuous in the summary of the results section (lines 275-278)

Finally I think a little more detail on the unique environment would be welcome.

---

## Round 0.3 · Minor Revisions

· Academic Editor

Minor Revisions

You've done a good job of improving the paper, but there are a couple things that still need to be fixed.

First, the reviewer's point about 9/13 not being significant is a red herring, which I tried to explain last time. If four consistent results are strongly supported, it doesn't matter that the fraction of populations showing your effect is not significantly different from 50:50. Think about it this way. Suppose you did one experiment and found a significant result. You wouldn't downplay the importance of the result because 1/1 is not significantly different from 50:50. So, please get rid of the discussion about 9/13 not being significant, and focus on the results that are strongly supported statistically.

Second, I'd like you to discuss the matter of limited molecular resolution a little more fully. You have explained well that there might be more populations than you've been able to detect, owing to limited molecular resolution. However, you haven't addressed (and you need to do this only briefly) that an alternative explanation for the apparent population expansions is that instead there was recently an increase in the number of (undetected) populations. Please see my first review for more details and for a reference on this.

So, please address these points, and then you should be done.

---

## Round 0.4 · Minor Revisions

· Academic Editor

Minor Revisions

You've done a great job on the paper, and it is very close to official acceptance.

There is only one extremely minor thing that needs to be changed, and I apologize for not noticing it earlier. The title starts "Common population expansions in coexisting..." I believe there has been a problem perhaps in translating from Spanish to English here. "Common" in this case will be read by most English-speaking readers as "frequent" or possibly as "ordinary," neither of which is what you mean. I think you mean "shared" population expansions. So, may I suggest this title:

"Population expansions shared among coexisting bacterial lineages are revealed by genetic evidence"

(I don't see any rules against using a full sentence for a title, and it will be more comprehensible this way.)

Since you'll have to re-submit your revision anyway, I'd like to suggest a few extremely minor changes:

Line 98. "and" should not be italicized.

Line 258. It should be: "defined by the FST statistic...." (Of course, ST should be subscripted, as you have it; I just can't do that in this editorial box.)

Line 276 has two consecutive periods.

I hope you agree with my suggested change for the title, or perhaps you'll find something better. What follows is the standard letter for minor revisions.

---

## Round 0.5 · accepted · Accept

· Academic Editor

Accept

This is very nice work! Thank you for all your patience while we worked this out.